# Welfare Implications of Border Carbon Adjustments on the Trade of Harvested Wood Products

**DOI:** 10.3390/ijerph20010790

**Published:** 2022-12-31

**Authors:** Xinxin Liao, Zhuo Ning

**Affiliations:** College of Economics and Management, Nanjing Forestry University, Nanjing 210037, China

**Keywords:** dynamic game, border carbon adjustments (BCAs), carbon tariffs, harvested wood products (HWPs), carbon leakage

## Abstract

Border carbon adjustments (BCAs) are designed to regulate carbon emissions and reduce carbon leakage. Thus far, BCAs are mainly applied to imported carbon-intensive products. On the other hand, harvested wood products (HWPs) are the extension of forest carbon stocks, whose changes affect a country’s carbon stock level. Nonetheless, the trade of HWPs also raises the problem of carbon leakage when their carbon stocks are exported, which can be partially solved by applying export BCAs. We construct a two-stage game model to analyze the strategy changes of the government and forestry companies under BCAs: the first stage is output competition in a Cournot game similar to the trade of HWPs between New Zealand and China; the second stage is the setting of the tax rate of BCAs by the country. We use the inverse solution method to derive the results of the game. Our results find that the government imposes BCAs on exports of HWPs when the carbon stock value exceeds a threshold. Moreover, the export BCAs on HWPs can effectively reduce the amount of HWPs exported. The results also show that BCAs diminish forestry exporters’ revenues and consumer surplus while having no significant detrimental impact on a country’s welfare. BCAs help include carbon stock values into HWPs’ prices and reduce carbon leakage, which is beneficial for climate change. Thus, exporting countries can maintain their welfare by implementing BCAs, and the forestry companies can respond by improving product quality, enhancing product uniqueness, and reducing production costs.

## 1. Introduction

Carbon emissions have accumulated since the Industrial Revolution and caused global warming. Therefore, countries are increasingly calling for carbon border adjustments (BCAs) to prevent carbon leakage. BCAs are tariffs imposed on imports and exports in cross-border trade to balance the cost of carbon reduction borne by importing and exporting countries [1]. For example, the US Trade Policy Agenda 2021 and Annual Report 2020, issued in March 2021, explicitly stated that BCAs would be considered for the agenda in the future. In addition, the EU planned to impose BCAs on imports from five major industries in 2026. BCAs impose import tariffs on products from countries that do not adopt carbon reduction measures to increase their costs, shifting the economic burden of emission reduction to non-emission-reducing countries [2]. Furthermore, the BCAs level the playing field for emission-intensive products by applying a domestic carbon price to carbon emissions from traded goods [3].

Notably, the intensification of climate change has also led to an increasing focus on the role of harvested wood products (HWPs) in climate change mitigation. Forestry is important in mitigating climate change [4,5,6,7,8,9]. Specifically, after a forest stand is harvested, a significant portion of the wood remains stored in HWPs, affecting a country’s GHG inventories under the Kyoto Protocol treaty [10,11,12]. Overall, an increase in the HWPs’ carbon stock is the removal of carbon from the atmosphere, while a decrease in the carbon stock is considered emissions [13]. Although carbon stocks of HWPs are not as large as forest carbon sinks, their continuous increase is a significant contribution to emission reductions on a global scale [14,15], and the carbon pool of HWPs has great carbon removal potential [16]. For example, managing all of Thailand’s teak forests could reduce emissions by up to 19.8% of its autonomous national contribution by 2030 [17].

However, the use of and trade in HWPs affects a country’s carbon pool of HWPs [18]. Specifically, HWPs’ trade changes a country’s carbon storage, thus affecting its emission reduction level [19,20,21]. For example, China’s CO_2_ emissions from trading in HWPs were 121.25 Mt in 2014 [22]. Furthermore, by the end of 2015, the total carbon stock of HWPs exported from New Zealand to China was 227,000 tons [23]. Moreover, carbon leakage from the forestry sector is also noticeable [24,25]. It is worth noting that carbon leakage will reduce the impact of the forestry sector in mitigating climate change [26].

In summary, HWPs serve as carbon sinks, and their trading affects a country’s carbon stock. Especially under the accounting of the stock change method, exporting HWPs increases the economic benefits of domestic companies while decreasing a country’s carbon stocks. Furthermore, the trade of HWPs is accompanied by carbon storage leakage, which weakens the mitigation effect of HWPs [27]. On the other hand, the consumption of HWPs is uneven, with major consuming countries accounting for 69% of global HWP’s carbon stocks [25]. For example, China and India are the largest consumers of wood carbon stocks in the Belt and Road region [28]. Therefore, it is necessary to reduce carbon storage leakage from HWPs.

In other words, HWP exporting countries must balance companies’ benefits and values of carbon stocks in exporting HWPs to optimize their welfare. Because the carbon stored in HWPs can affect national greenhouse gas inventories [10,11,12], and carbon storage is given some value in the carbon market instead of the cost of carbon emissions, the value of carbon storage in HWPs can be measured in economic terms.

The paper assumes that the exporting country of HWPs implements the border adjustment, i.e., the government levies taxes on exported HWPs to compensate for the loss of carbon stocks and reduce carbon leakage. On the other hand, the government can provide export subsidies to the companies to promote the forestry industry if it is more economically beneficial. Both export tariffs and subsidies are BCAs, and their form depends on the measurement of carbon stock value and economic benefits from the forestry industry. Specifically, if the economic benefits of the forestry companies are dominant, the government prefers to implement export subsidies; if the carbon stocks value is dominant, the government prefers to apply export tariffs.

Therefore, based on the importance of HWPs for climate change mitigation and the trend of implementing BCAs, this paper uses BCAs to address the carbon leakage from HWPs. We developed a game model to analyze the behavioral choices of governments and companies in scenarios with and without BCAs. The aim is to explore the feasibility of implementing BCAs on the export of HWPs to reduce a country’s carbon stock leakage and to investigate the impact of BCAs on countries’ welfare. The analysis will answer two specific questions: 

(1) Can BCAs for HWPs reduce carbon stock leakage while maintaining a country’s welfare?

(2) What actions should forestry companies take under BCAs? 

Finally, according to the results of this paper, we provide specific suggestions to trading countries and companies, respectively.

## 2. Literature Review

Thus far, the discussion on BCAs has mainly focused on import carbon tariffs imposed on carbon-intensive products and their economic and environmental impacts. Regarding the environment, some scholars point out that implementing BCAs limits carbon emissions and leakage [29,30,31,32], but is also expensive [33]. In particular, Zhang et al. (2020) compared BCAs with general tariffs and found that BCAs were more effective in reducing carbon leakage [34]. However, some other scholars point out that BCAs have limited effects on carbon leakage. For example, Fang et al. (2020) analyzed a global supply chain model and found that carbon tariffs do not necessarily reduce global emissions [35].

Regarding economics, BCAs may reduce trade flows and lower a country’s welfare [33,36]. Sutherland (2020) notes that the cost burden of BCAs is heaviest for developing and industrialized countries [37]. For example, if an import tariff of $50 per ton of carbon were imposed, imports from China, India, and South Africa would face an average tariff rate of 10%, 8%, and 12%, respectively [38]. Moreover, an anti-trade scenario with an additional 25% tariff would reduce global export volumes by 32.5% [6]. Sheng and Wang (2022) also show the negative impacts of carbon tariffs imposed by developed countries on China’s economy [39]. Furthermore, high carbon tariffs would reduce carbon-intensive products’ competitiveness in foreign markets [32,40]. However, Hong et al. (2022) show that adopting BCAs would induce other countries to raise their domestic emission tax rates, so excessive shrinkage of domestic production may not happen [41]. Mörsdorf (2022) claims that BCAs would generate significant revenues for governments, which could be used to support low-carbon innovation and mitigation across countries [42].

Different BCAs often lead to different behaviors from companies. Moreover, companies’ strategies are an important factor affecting the impact of BCAs, so studying these strategies is a crucial part of the research on BCAs. Moreover, game theory can analyze strategies’ impact on the outcome and allows for exploration of interplay between game participants [43]. Therefore, game theory is a popular research approach to model the strategic choices of companies and governments. For example, Eyland and Zaccour (2014) explore the impact of BCAs in bilateral trade with different carbon taxes by establishing a three-stage game model [30].

Furthermore, game theory is often used to explore participants’ behavior and the impact of welfare policies. Wang et al. (2016) built a two-stage game to investigate textile firms’ strategies under carbon tariffs in developing countries [44]. Hou and Jia (2016) use a three-stage game to study the government’s and enterprises’ optimal choices under BCAs [45]; Fang et al. (2020) use game theory to analyze the global abatement effects of BCAs [35]. In addition, Huang et al. (2022) used a game to analyze firms’ responses to BCAs and made relevant suggestions for companies, such as cost reduction [46]. Finally, Rustico and Dimitrov (2022) analyze the impact of a carbon tax on corporations and welfare by comparing the game in different states [47].

The research objects of the existing literature are mainly carbon-intensive products. However, besides those products, we also need to consider carbon storage products as HWPs, which impact a country’s carbon pool. Based on the incomprehensiveness of the current research objectives and the significant role of HWPs in climate change mitigation, we believe that carbon border regulation for HWPs is worth studying. However, few studies address the issue of carbon leakage from HWPs. García et al. (2018) pointed out that governmental management policies are crucial to influencing carbon leakage in the forestry sector, but their research does not hold a trade perspective [48]. Zhang et al. (2020) state that reflecting the carbon stock value of forest products in their prices can effectively address carbon leakage, but does not indicate the specific form [49]. Peng and Ning (2021) proposed BCAs to solve the carbon leakage problem of HWPs, but did not analyze the impact empirically [50].

Therefore, the current research on BCAs mainly focuses on their environmental and economic impacts, which vary under different research objects and methods. Moreover, although some scholars point out the utilization of BCAs to solve carbon leakage from HWPs, limited research has analyzed the specific impact.

## 3. Methods and Basic Assumptions

The basic model framework is referenced from Patel’s study on dynamic games in international trade [51]. Figure 1 shows the structure of our model, and all adopted parameters and their definitions are listed in Table 1.

This paper simulates unilateral trade to focus on the export segment, such as log exports from New Zealand to China [30,45,52]. Moreover, to explore the changes in countries’ carbon stock in international trade, this paper assumes the stock change method to account for the countries’ carbon stocks. Under this accounting method, the export of HWPs is regarded as carbon emissions by the exporting country; conversely, the importing country increases its carbon stocks [53]. The carbon stocks of HWP are affected by multiple factors [54,55], so it is assumed that the amount of carbon storage per unit of HWPs is denoted by the parameter m (m>0), and the constant parameter ε denotes the value of carbon storage per unit of HWP.

Our game model considers two countries with open economies: Country *E* and Country *I*, where Country *E* is the producer and exporter of HWPs, and Country *I* is the importer of HWPs of country *E*. As in the case of New Zealand’s HWP trade with China, the amount of HWPs consumed domestically in Country *E* is negligible compared to the amount exported. Therefore, we ignore the domestic market of Country *E* and assume that Country *I* is the only market for the HWPs in the game. Moreover, Country *I* can also produce homogeneous HWPs except by importing. We assume that *I* and *E* have only one company, Company *i* and Company *e*, respectively, based on Wang et al. (2016) and Hecht and Peters (2019) [44,56]. Company *e*’s output is for export only, because we ignore the market in Country *E* [44,56].

Regarding the game model, we assume that the two companies are engaged in a Cournot game, in which the two players, Company *e* and Company *i*, produce and sell the same product in the market. Their marginal production costs are $C_{i}$ and $C_{e}$, and the demand curve of the market they jointly face is linear, i.e., the market has a uniform market price. In the Cournot model, the demand for the commodity is linear. We also assume that the demand function for the HWPs in Country *I* is $p=\alpha -\beta \left(Q_{i}+Q_{e}\right)\;(\alpha ,\beta >0)$, where *p* denotes the price of HWPs, $Q_{x}$ is the output of Company x (x=e or i), α denotes the limiting price of the HWPs when the output is 0, and β denotes the sensitivity of price to supply. Assume that the marginal cost of producing HWPs in Company x is $C_{x}$ (x=e or i). Referring to Hecht and Peters’ model [56], we also assume that Company *e* is a large producer of HWPs and has lower production costs, which means $C_{i}>C_{e}>0$. For simplicity, transportation costs are assumed to be 0. The limiting price must be greater than the marginal cost, because a rational producer will always produce when the price is greater than the marginal cost, so $\alpha >C_{i}>C_{e}$.

Furthermore, this paper compares two scenarios: with and without BCAs. We assume that the game in both scenarios is played under the condition of perfect information. In the scenario without BCAs, the only players involved in the game are Companies *i* and *e*, whose strategic choice is their output to maximize their profits, which is equivalent to oligopolistic competition. We use $\pi_{x}$ to denote the profit of Company *x* (*x* = *e* or *i*). The scenario with BCAs is a two-stage game involving Country *E*, Company *i,* and Company *e*. In the first stage of the game, Country *E* determines the form of BCAs as export carbon tariffs or subsidies to maximize a country’s welfare—the exporting country can impose tariffs to make countries that import HWPs pay for the value of carbon stocks in HWPs or give export subsidies to exporters in order to promote the development of its forest industry. We denote the rate of BCAs by *t* and $W_{X}$ denotes the welfare of Country *X* (*X* = *E* or *I*). The sign of *t* indicates whether it is an export subsidy or a tariff—a negative *t* means giving an export subsidy, and a positive *t* means imposing a tariff. In the second stage, Companies *i* and *e* simultaneously choose their outputs to maximize profits. We use the superscript *N* to denote the equilibrium value of the scenario without BCAs and the superscript *B* to denote the equilibrium value with BCAs.

Our game is a two-stage game. Similar to other studies, in solving the two-stage game, we use the inverse solution method [44,56,57,58,59,60,61]. The method’s basic idea is to start from the last stage of the dynamic game and determine the optimal action of the participants by maximizing their interests in their historical conditions. The reason is that, in a complete and perfect dynamic game, the first rational player inevitably considers how the latter player would act in the later stage. For a two-stage game, the player can make an informed choice in the second stage of our game, i.e., when there is no longer a subsequent stage to hold the choice. After the strategy chosen by the game player in the later stage is determined, it is easy for the player in the previous stage to choose a strategy. The reverse induction method analyzes the dynamic game’s last stage and gradually summarizes each stage’s strategy choice. Thus, in this paper, we first solve for the output that maximizes the benefits of Companies *e* and *i* and then bring this output into the welfare function of Country *E* to find the BCA level that maximizes its value.

## 4. Game Process and Results

### 4.1. Game without BCAs

In the scenario without BCAs, Companies *e* and *i* choose their outputs to maximize their benefits, whose profit functions include their income and costs, as shown by Equation (1). We determine the optimal strategies of the companies to maximize their profit functions:(1)$$\{\begin{array}{c} \underset{Q_{e}}{\max}\pi_{e}\left(Q_{e},Q_{i}\right)=\left(\alpha -\beta Q_{e}-\beta Q_{i}\right)Q_{e}-C_{e}Q_{e}\\ \underset{Q_{i}}{\max}\pi_{i}\left(Q_{e},Q_{i}\right)=\left(\alpha -\beta Q_{e}-\beta Q_{i}\right)Q_{i}-C_{i}Q_{i} \end{array}$$

The optimal outputs of Companies *e* and *i* can be obtained by differentiating the above equations and making them equal 0. We use the superscript *N* to denote the equilibrium value of the scenario without BCAs. Equation (2) presents the optimal output for Companies *e* and *i* without the BCAs scenario:(2)$$\{\begin{array}{c} Q_{e}^{N}=\frac{\alpha +C_{i}-2C_{e}}{3\beta }\\ Q_{i}^{N}=\frac{\alpha +C_{e}-2C_{i}}{3\beta } \end{array}$$

Bringing the companies’ optimal outputs into the price function, we obtain the equilibrium price of HWPs as:(3)$$p^{N}=\alpha -\beta \left(Q_{e}^{N}+Q_{i}^{N}\right)=\frac{\alpha +C_{i}+C_{e}}{3}$$

Then we bring the optimal outputs into the profit function, and we can find the respective profits of companies *e* and *i* as:(4)$$\{\begin{array}{c} \pi_{e}^{N}\left(Q_{e},Q_{i}\right)=\frac{{\left(\alpha +C_{i}-2C_{e}\right)}^{2}}{9\beta }\\ \pi_{i}^{N}\left(Q_{e},Q_{i}\right)=\frac{{\left(\alpha +C_{e}-2C_{i}\right)}^{2}}{9\beta } \end{array}$$

Next, we calculate countries’ welfare. Since we ignore the HWPs market in Country *E*, Country *E*’s welfare consists of Company *e*’s profits ($\pi_{e}^{N}$) and the country’s carbon stock value. On the other hand, Country *I*’s welfare function has three components: Company *i*’s profits *(*$\pi_{i}^{N}$); the country’s carbon stock value ($\varepsilon Q_{e}^{N}\left(t\right))$; and the consumer surplus (CS). Under the stock change method, Company *e*’s exporting of HWPs reduces Country *E*’s carbon stocks; on the contrary, the country importing HWPs increases. Therefore, we can get the welfare of Countries A and B, respectively, as:(5)$$\{\begin{array}{l} W_{E}^{N}=\pi_{e}^{N}\left(Q_{e}^{N}\left(t\right),Q_{i}^{N}\left(t\right)\right)-\varepsilon Q_{e}^{N}\left(t\right)=\frac{\left(\alpha +C_{i}-2C_{e}\right)\left(\alpha +C_{i}-2C_{e}-3\varepsilon \right)}{9\beta }\\ W_{I}^{N}=\pi_{i}^{N}\left(Q_{e}^{N}\left(t\right),Q_{i}^{N}\left(t\right)\right)+CS+\varepsilon Q_{e}^{N}\left(t\right)=\frac{{\left(\alpha +C_{e}-2C_{i}\right)}^{2}}{9\beta }+\frac{{\left(2\alpha -C_{e}-C_{i}\right)}^{2}}{18\beta }+\varepsilon \frac{\alpha +C_{i}-2C_{e}}{3\beta } \end{array}$$

### 4.2. Game with BCAs

The game with BCAs is a two-stage game. In the first stage, Country *E* determines the form and tax rate of BCAs; in the second stage, Company *e* and Company *i* determine their outputs.

As in other studies, we use the inverse solution method to solve the game. Therefore, we first consider the second stage. In this scenario, the profit function of Company *e* is composed of three parts: income, production cost, and export carbon tariff or subsidy. As *m* denotes carbon stocks in a unit of HWPs, exporting $Q_{e}$ units of HWPs are subject to export tariffs or subsidies of $mtQ_{e}$. It should be noted that a positive *t* represents the imposition of the export tariff, and a negative value of *t* represents the export subsidies. Therefore, the profit function of Company *i* is composed of two parts: income and production cost. Note that Country *E* is the exporter of HWPs, which imposes BCAs on exported HWPs, and Country *I*’s products are for domestic consumption only, whose HWPs are not subject to BCAs. Thus, Company *e* has an additional carbon border adjustment tax in the profit function, compared to Company *i*’s. The profit maximization problem for the two companies can be expressed as follows:(6)$$\{\begin{array}{c} \underset{Q_{e}}{\max}\pi_{e}\left(Q_{e},Q_{i}\right)=\left(\alpha -\beta Q_{e}-\beta Q_{i}\right)Q_{e}-C_{e}Q_{e}-mtQ_{e}\\ \underset{Q_{i}}{\max}\pi_{i}\left(Q_{e},Q_{i}\right)=\left(\alpha -\beta Q_{e}-\beta Q_{i}\right)Q_{i}-C_{i}Q_{i} \end{array}$$

We use the superscript *B* to denote the equilibrium value in the scenario with BCAs. Then, solving for the first-order derivative of Equation (6), we obtain the optimal outputs of Companies *e* and *i* with BCAs scenario, respectively:(7)$$\{\begin{array}{l} Q_{e}^{B}\left(t\right)=\left(\alpha +C_{i}-2C_{e}-2mt\right)/\left(3\beta \right)\\ Q_{i}^{B}\left(t\right)=\left(\alpha -2C_{i}+C_{e}+mt\right)/\left(3\beta \right) \end{array}$$

The total output and the price of HWPs for the two companies are:(8)$$Q_{e}^{B}\left(t\right)+Q_{i}^{B}\left(t\right)=\frac{2\alpha -C_{i}-C_{e}-mt}{3\beta }$$
(9)$$p^{B}=\alpha -\beta \left(Q_{e}^{B}\left(t\right)+Q_{i}^{B}\left(t\right)\right)=\frac{\alpha +C_{i}+C_{e}+mt}{3}$$

Under the assumption *t* > 0, the government imposes export carbon tariffs on companies. From a simple static comparison to Equation (7), an increase in the export carbon tariff leads to a decrease in the export volume of Company *e* and an increase in the output of Company *i*. This is because, in the Cournot game, the cost of the product determines the sales volume—the lower the cost of the product, the higher the sales volume. Therefore, the BCAs imposed on Company *e* increase its cost, while the relative cost of Company *i* is reduced, whose cost is not affected by the BCAs. Furthermore, we can see from Equations (8) and (9) that the total amount of HWPs in the market of Country *I* decreases with the increase of carbon tariffs, while the price increases with carbon tariffs simultaneously. The reason is that the export carbon tariffs increase the cost of Company *e*, reducing its competitive advantage and sales volume and making Company *i* more competitive. Further, the decrease in Company *e*’s supply leads to a decrease in the market’s supply and an increase in the price.

In addition, we are comparing Equations (3) and (9) and find that the carbon stock can be reflected in the price of HWPs under the BCAs scenario, which means BCAs make the price of HWPs respond to the carbon stock of HWPs and likewise raise the price of HWPs. Furthermore, after calculation, the tariff is borne two-thirds by Company *e* and one-third by the consumers.

Then bringing Equation (7) into Equation (6), we can obtain the profits of Companies *e* and *i*, respectively, as follows:(10)$$\{\begin{array}{c} \pi_{e}^{B}=\frac{{\left(\alpha +C_{i}-2C_{e}-2mt\right)}^{2}}{9\beta }\\ \pi_{i}^{B}=\frac{{\left(\alpha -2C_{i}+C_{e}+mt\right)}^{2}}{9\beta } \end{array}$$

By analyzing Equation (10), it is known that the imposition of carbon tariffs reduces the profit of export companies, while Company *i*’s profit increases as the carbon tariff increases. The reason is that Country *E* raises its export carbon tariff to prevent carbon leakage, which raises the company’s marginal cost and reduces its competitiveness, resulting in lower sales volume and profits. In contrast, Company *i* becomes more competitive, and its sales volume and profits increase.

For the first stage, Country *E* determines the form of BCAs and the tax rate to maximize its welfare. Based on the assumption, Country *E*’s objective function can be expressed as:(11)$$\underset{t}{\max}W_{E}^{B}\left(t\right)=\pi_{e}^{B}\left(Q_{e}^{B}\left(t\right),Q_{i}^{B}\left(t\right)\right)+tmQ_{e}^{B}\left(t\right)-\varepsilon Q_{e}^{B}\left(t\right)$$

Bringing from Equation (7) in Equation (11), we get Country *E*’s welfare as:(12)$$W_{E}^{B}\left(t\right)=\frac{\left(\alpha +C_{i}-2C_{e}-3\varepsilon +mt\right)\left(\alpha +C_{i}-2C_{e}-2mt\right)}{9\beta }$$

Take the derivative to t for Equation (12), and let the equation $W_{E}^{B^{\prime }}\left(t\right)=0,$ we get the optimal strategy for BCA as:(13)$$t^{B}=\frac{-\alpha -C_{i}+2C_{e}+6\varepsilon }{4m}$$

Then we solved the quadratic function of Equation (12) about t, and the result is $W_{E}^{B^{\prime \prime }}\left(t\right)=-4m^{2}<0$, so the value of t must be a maximum value. Equation (13) shows that the sign of $t^{B}$ is ambiguous, and the larger the carbon stock value ε is, the larger the rate of carbon tariff. It implies that the larger the value of HWP, the more Country *E* wants to keep the HWPs within the country to maintain its carbon stocks. Therefore, Country *E* reduces its export volume by a higher carbon tariff rate. If the value of ε is small, the carbon tariff can be negative, interpreted as export subsidies from Country *E* to Company *e*.

Bringing Equation (14) into Equations (6) and (11), we can get the welfare of the two countries and the profits of the two companies as:(14)$$\{\begin{array}{c} \pi_{e}^{B}=\frac{{\left(\alpha +C_{i}-2C_{e}-2mt\right)}^{2}}{9\beta }=\frac{{\left(\alpha +C_{i}-2C_{e}-2\varepsilon \right)}^{2}}{4\beta }\\ \pi_{i}^{B}=\frac{{\left(\alpha -2C_{i}+C_{e}+mt\right)}^{2}}{9\beta }=\frac{{\left(\alpha -3C_{i}+2C_{e}+2\varepsilon \right)}^{2}}{16\beta } \end{array}$$
(15)$$\{\begin{array}{l} W_{E}^{B}\left(t\right)=\frac{{\left(\alpha +C_{i}-2C_{e}-2\varepsilon \right)}^{2}}{8\beta }\\ W_{I}^{B}\left(t\right)=\frac{{\left(\alpha -3C_{i}+2C_{e}+2\varepsilon \right)}^{2}}{16\beta }+\frac{{\left(3\alpha -C_{i}-2C_{e}-2\varepsilon \right)}^{2}}{32\beta }+\frac{\varepsilon \left(3\alpha -C_{i}-2C_{e}-2\varepsilon \right)}{4\beta } \end{array}$$

Bringing $t^{B}$ into Equations (8)–(10) and (12), we obtain Table 2, which presents and compares the differences between the welfare functions of Countries *E* and *I* and the profit functions of Companies *e* and *i* in the two scenarios.

By comparing the two scenarios, we find that the profit of Company *i* monotonously increases when Country *E* imposes export carbon tariffs in the BCAs scenario. At the same time, Company *e*’s profit and Country *E*’s welfare decrease with higher carbon stock value. The decrease in Company *e*’s profit is because the carbon tariff increases its marginal cost, while the decrease in Country *E*’s welfare results from the double effects of Company *e*’s decreasing profit and the loss of carbon stock of HWPs. In order to demonstrate the game results more intuitively, numerical simulations are carried out in the next section.

### 4.3. Numerical Analysis

The game’s parameters will be assigned in this section to explore better the influence of BCAs on countries’ welfare and companies’ profitability. Referring to existing literature studies and the trade of HWPs, the parameters for this paper are set as follows [30,44].

We bring the parameters in Table 1 to the equations to obtain the game’s results. Figure 1 presents the trend of the BCA rate with the values of carbon stocks.

From Figure 2, when the value of carbon stock is less than 15, the BCA imposed by Country *E* is negative, indicating subsidies for exported HWPs. On the other hand, when the carbon stock value ε is greater than 15, Country *E* imposes export carbon tariffs on exported HWPs. It implies that when the carbon stock value is small, the economic benefit gained by Company *e* outweighs the loss of carbon stocks of HWPs for Country *E*. Therefore, the country gives export subsidies, providing Company *e* with greater economic benefits and thus enhancing the welfare level of Country *E*. However, as the value of carbon stocks continues to rise, the economic benefit gained by Company *e* is not significant enough to make up for the lost value of carbon stocks for Country *E*, so it imposes export carbon tariffs on the HWPs to make the importing country pay for carbon storage in HWPs.

Next, we focus on the variation of Companies’ profits with BCAs. Figure 3 shows that Company *e*’s profits decrease as the BCAs increase. When the export subsidy is applied to Company *e*, it earns more profits. However, when export carbon tariffs are imposed, Company e’s profits decrease as the carbon tariff rate increases. In contrast, Company *i*’s profits move in the opposite direction from Company *e*’s, which implies that if the government imposes carbon tariffs on exports, the competitiveness of the exporters will suffer. Figure 4 shows the change in consumer surplus similar to that of Company *e*—both decrease as the BCAs tax rate increases. The reason is that export carbon tariffs increase the price of HWPs, thereby reducing the consumer surplus.

Finally, we explore the impact of BCAs on a country’s welfare. Figure 5 shows that the welfare of Countries *E* and *I* is higher when BCAs are implemented. The reason is that the border regulation in Country *E* can balance the economic benefits and the carbon HWPs’ stock value to maximize its welfare. For Country *I*, which enjoys the benefits from the adjustment of Country *E*, the lower HWP prices lead to an increase in consumer surplus in Country *I* when Country *E* subsidizes Company *e*’s exports; and Company *e*’s impaired competitiveness by being imposed export tariffs leads to an increase in Company *i*’s profits. This result suggests that the government’s border regulation behavior is justified to maintain welfare for exporting and importing countries.

The export carbon tariff does not harm the exporting country’s welfare, but reduces the exporting company’s profits and consumer surplus. Specifically, BCAs include the value of carbon stocks in HWPs in the price, making countries that buy HWPs pay for the value of carbon stocks. We infer from Figure 3 and Figure 4 that the exporting company and consumers in the importing country share the carbon tariff cost. Compared with consumers, Company *e* has a better chance to improve its profits. Thus, we adjust the parameters to explore the companies’ responses to maintain their profits. Figure 6, Figure 7 and Figure 8 present the effects of price elasticities of supply, production costs, and carbon stocks per unit of HWPs on the exporters’ profit, respectively.

Figure 6 shows that the smaller the elasticity, the greater the Company *e*’s profit. It implies that Company *e* can reduce its supply elasticity by improving the quality and uniqueness of its products to maintain its profits at a higher level. Nevertheless, no matter how the elasticity changes, the curves’ trend does not change significantly, which means that the change in elasticity does not affect the impact of BCAs on the company’s profit. Similarly, Figure 7 shows that reducing the production cost also enhances company profits. Therefore, companies can increase their profits by improving production efficiency to reduce costs. Additionally, production costs do not affect the impact of BCAs on company profits.

The impact of HWPs’ carbon storage on Company *e*’s profits is more complex, as shown in Figure 8. The larger *m* is, the more the company’s profits are affected by BCAs, which are policies that target carbon stocks in HWPs. Using t = 0 as a threshold, when t < 0, the larger the carbon stock of HWPs, the larger the company’s profits; when t > 0, the result is reversed. This trend indicates that carbon stocks in HWPs affect companies’ profits, but it is difficult for companies to adjust the amount of carbon stored in HWPs.

## 5. Discussion

This paper discusses the impacts of BCAs on countries’ welfare and companies’ profits using a dynamic game and comparative static analysis with and without BCAs. With the two-game scenarios, we numerically simulated the final game outcome and the company’s response and obtained the following main results:

(i) The carbon stock value of HWPs is the main factor influencing a country’s decision on BCAs.

(ii) Imposing BCAs can reduce carbon stock leakage from HWPs and will not affect the welfare of the HWP exporting countries.

(iii) Under BCAs, companies can enhance their profits by improving product quality and uniqueness and reducing production costs or lean manufacturing.

Result (i) shows that carbon stock values influence levels of BCAs—exporting countries impose carbon tariffs on exported HWPs only when the carbon stock value is greater than the threshold, while the government grants export subsidies to companies when the carbon stock value is lower than the threshold. The same conclusion is also found in Eyland and Zaccour (2014), which concludes that the government chooses to implement BCAs to maximize its benefits only if a certain threshold is breached [30].

For result (ii), imposing export carbon tariffs on HWPs would not hurt countries’ welfare. For the HWPs’ exporters, carbon tariffs reduce their companies’ profits, but do not harm the country’s welfare. For the country importing HWPs, the other country’s export carbon tariffs make their import more expensive and less supplied but increase their companies’ profits, so they do not negatively affect the country’s welfare. This conclusion contrasts those drawn for carbon-intensive products, which indicated that BCAs negatively impacted countries’ welfare [2,39]. The reason for this is that, unlike punitive BCAs for carbon-intensive products, BCAs on HWPs are designed to balance the economic benefits and carbon stock values and are aimed at making carbon stock importing countries pay for carbon stocks.

On the other hand, the results of this paper on welfare are close to Böhringer et al. (2017), in which BCAs may increase the country’s welfare [62]. As a result, while HWPs exporters have lower economic gains from the imposition of BCAs, they also reduce their carbon stock losses, which means they have more carbon assets. In addition, we found that BCAs are beneficial for climate change. Paluš et al. (2020) show that restricting the export of some HWPs can help enhance the mitigation effect of HWPs on climate change [63]. Furthermore, BCAs make the price of HWPs reflect the carbon stock values in HWPs. As Zhang et al. (2020) once suggested, assigning carbon storage values to HWPs would increase HWPs’ prices and is vital for mitigating climate change [49].

The imposition of carbon tariffs on exported HWPs has resulted in higher prices for HWPs and lower consumer demand. As in result (iii), forest companies reduce the production and export of HWPs, affecting consumer surplus and the company’s profits. However, this result serves as a reference for forestry companies to reduce costs and increase efficiency under BCAs, which is mentioned by Huang et al. (2022) [46].

Therefore, based on the results of this paper and current policy preferences, it is feasible for large exporting countries of HWPs to maintain their welfare through BCAs. However, BCAs increase the cost of HWP importers, who have a few HWP suppliers mostly. For example, China’s major HWP importers are the Russian Federation, New Zealand, Germany, the United States of America, and Vietnam, which account for 16.72%, 14.11%, 9.2%, 8.11%, and 6.12% of China’s imports of HWPs in 2021, respectively. In addition, according to China’s customs data, China imported 290 million m^3^ of the wood equivalent of various HWPs in 2019, accounting for 54.37% of the total supply of its HWPs market. The dependence on international timber supply has left China vulnerable to the risks arising from price fluctuations in international HWPs [64]. To conclude, export tariffs on HWPs would benefit exporting countries’ welfare, but countries who are chronically dependent on importing HWPs will be at risk.

Like BCAs, many countries have already adopted export control to better utilize their HWPs—for example, Russian export tariffs for some HWPs [65]. In turn, many HWP-importing countries reduced import tariffs on HWPs to ensure the availability of their own HWPs, e.g., China announced a 50% reduction in tariffs on some American imports of HWPs in 2020 [66]. The above trends indicate countries’ willingness to keep more HWPs domestically. However, there is more than one supplier of HWPs, and decisions between exporting countries can affect each other. As Keen et al. (2022) show, the BCA implementation of a country is influenced by others [67]. For example, if Russia does not implement BCAs, New Zealand should get ready for export reduction to China before it decides to implement them.

Inconsistently with reality, there are only two participants in our game, which models that all exporters value carbon storage of HWPs and implement BCAs as they are united when the study is expanded to multiple trading partners. Currently, BCAs are the decision of developed countries—for example, the EU’s Carbon Border Adjustment Mechanism and the US Trade Policy Agenda 2021. In the future, as more countries value HWPs’ carbon storage, more countries will implement BCAs, so the real story will likely be this paper’s equilibrium state.

## 6. Conclusions

The conclusions of this paper are as follows: 

(1) BCAs on HWPs can reduce carbon stock leakage from HWPs.

(2) The welfare of exporting countries is not negatively affected by levying BCAs. 

(3) Forestry companies can respond to BCAs by improving product quality, enhancing the uniqueness of their products, and reducing the costs.

The findings of this paper provide comprehensive lessons for HWP trading partners to maintain their welfare. Based on the results, export countries can maintain their welfare by imposing BCAs and reducing carbon leakage, especially when the implementation is a joint action of most exporters. Therefore, if international organizations provide more information on the carbon stock value of HWPs and help include the measurements in some carbon accounts, more countries will adopt BCAs as a union, and the welfare of those countries can maintain. On the other hand, many international organizations have highlighted the importance of carbon stocks in HWPs in their documents, such as the Intergovernmental Panel on Climate Change (IPCC) and the Food and Agricultural Organization (FAO). Specifically, IPCC provides guidance on how to estimate and report annual carbon removals from HWPs, and the FAO document describes the methodology for carbon stock accounting in HWPs and their contribution to climate change [68]. These have inspired governments to take measures to maintain their country’s carbon stocks of HWPs.

On the other hand, from the perspective of the importing countries, even if the exporting countries do not act as a union but impose BCA independently, it is difficult for the importing countries to find alternative HWP suppliers to make up for the short term. Thus, each HWP importing country should expand its suppliers, to reduce the dependence on individual countries.

For companies, improving product quality, enhancing product uniqueness, and reducing production costs can reduce the cost of HWPs and increase their profits, which have higher requirements on companies’ production processes. Thus, the government can subsidize forestry companies to improve their production technologies. Moreover, importing countries must use their domestic forest resources efficiently to maintain their consumer surplus. At the same time, to reduce dependence on specific countries, importing countries can broaden their suppliers of HWPs.

This paper is an ideal game that involves only two countries and only one company per country. Therefore, future research can include more participants, to explore the effect of implementing BCAs policies. Furthermore, this paper does not consider carbon emissions in producing HWPs. Therefore, further research can consider the carbon emissions in the production of HWPs in the game and link carbon tax and carbon tariffs to investigate forestry companies’ more realistic commercial strategies.

## Figures and Tables

**Figure 1 ijerph-20-00790-f001:**
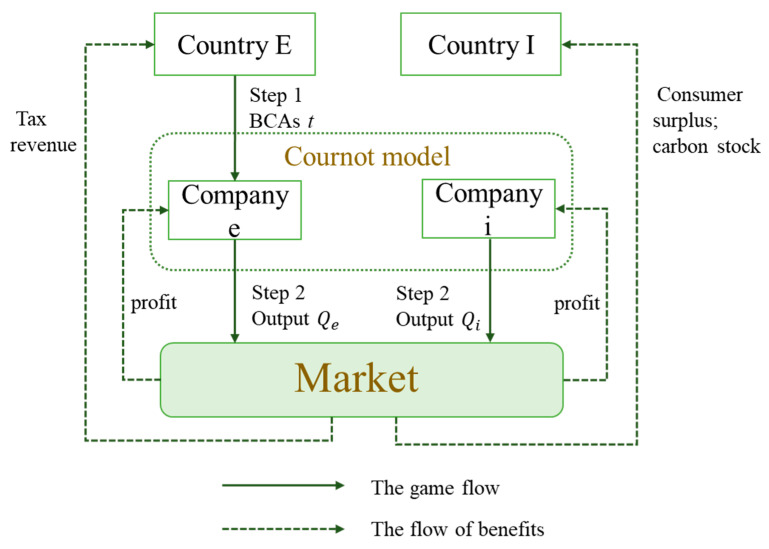
Framework of the game model. t denotes the tax rate of BCAs levied by Country *E*; $Q_{x}$ denotes the output of Company x (x=i or e).

**Figure 2 ijerph-20-00790-f002:**
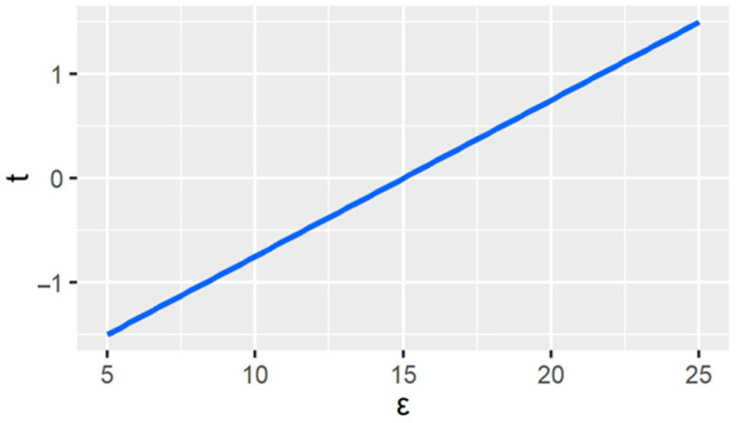
Variation of border carbon adjustments (BCAs) with carbon stock value of harvested wood products (HWPs). ε represents the value of carbon storage per unit HWPs, and t represents BCAs’ tax rates.

**Figure 3 ijerph-20-00790-f003:**
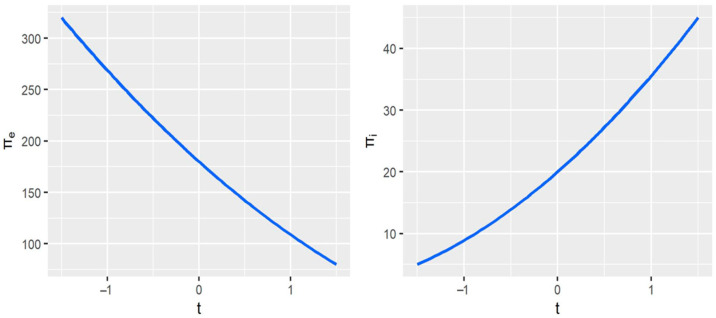
Movement of companies’ profits with border carbon adjustments (BCAs). *t* represents BCAs’ tax rates, and $\pi_{x}\left(x=e,i\right)$ represents the profit of the Company x (x=e or i).

**Figure 4 ijerph-20-00790-f004:**
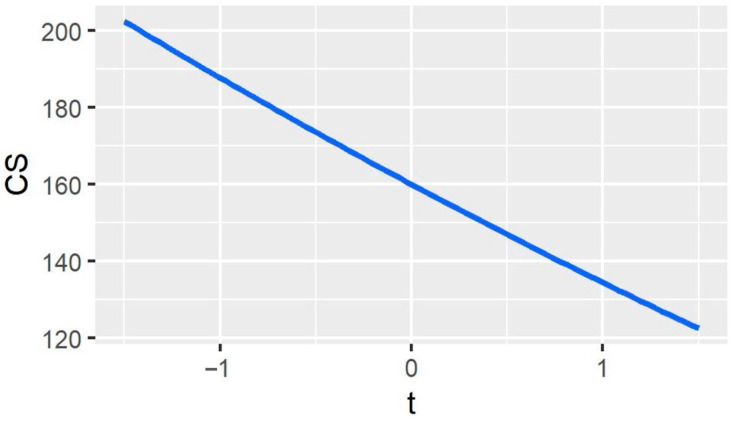
Movement of consumer surplus with border carbon adjustments (BCAs). *t* represents BCAs’ tax rates, and CS represents the consumer surplus of the Company i.

**Figure 5 ijerph-20-00790-f005:**
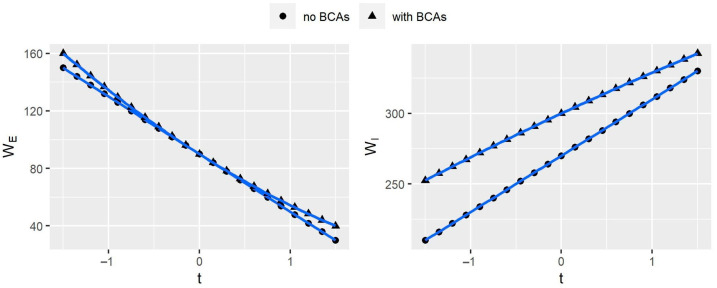
Movement of countries’ welfare with border carbon adjustments (BCAs). *t* represents BCAs’ tax rates; $W_{i}\left(i=I,E\right)$ represents the welfare of Country i (i=I or E).

**Figure 6 ijerph-20-00790-f006:**
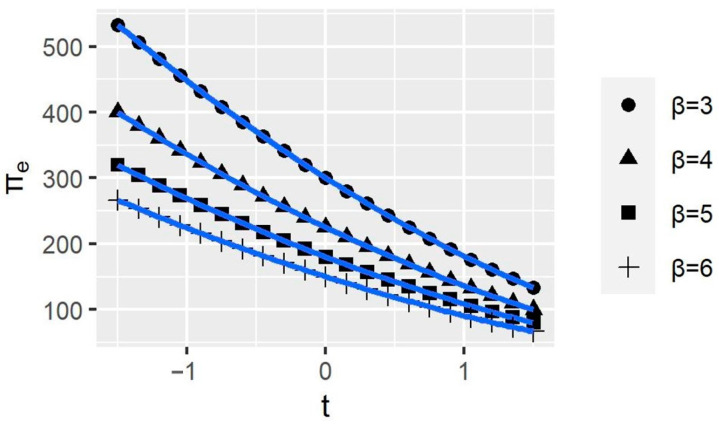
The variations of the company profits with different supply elasticities. *t* represents BCAs’ tax rates; $\pi_{e}$ represents the profit of Company e; and β represents HWPs prices to supply.

**Figure 7 ijerph-20-00790-f007:**
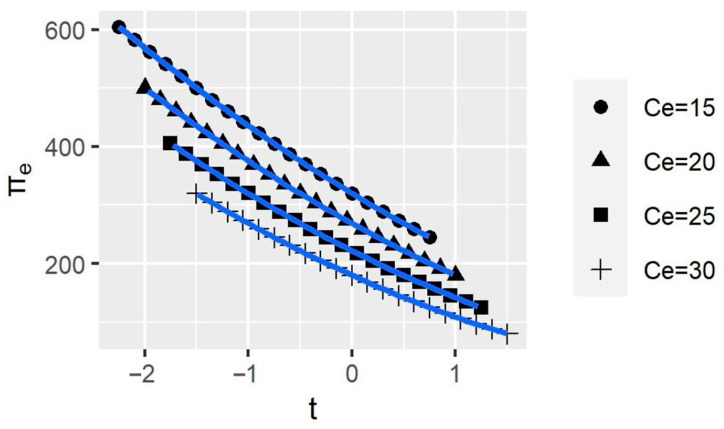
The variation of the company profits with different marginal costs. *t* represents BCAs’ tax rates; $\pi_{e}$ represents the profit of Company e; and $C_{e}$ represents the marginal production cost of Company e.

**Figure 8 ijerph-20-00790-f008:**
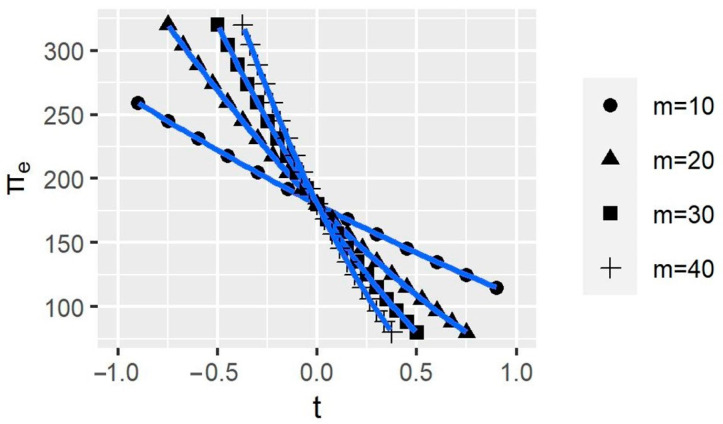
The variations of the company profits with different carbon storage in unit harvested wood products (HWPs). *t* represents BCAs’ tax rates; $\pi_{e}$ represents the profit of Company e; and *m* represents carbon storage in one unit of HWPs.

**Table 1 ijerph-20-00790-t001:** Parameter definitions and values.

Parameter	Definition	Value
α	The limit price when HWPs’ supply tends to zero	100
β	Sensitivity of HWPs prices to supply	5
$C_{e}$	The production cost of Company *e*	30
$C_{i}$	The production cost of Company *i*	50
ε	Carbon storage value	5~25
m	Carbon storage in unit HWPs	10
p	The price of HWPs	––––
$Q_{e}$	The output of Company e	––––
$Q_{i}$	The output of Company i	––––
$\pi_{e}$	The profit of Company e	––––
$\pi_{i}$	The profit of Company i	––––
$W_{E}$	The welfare of Country *E*	––––
$W_{I}$	The welfare of Country *I*	––––
CS	The consumer surplus	––––

**Table 2 ijerph-20-00790-t002:** Comparison of game results.

	Without BCAs	With BCAs
Price	$\frac{\alpha +C_{i}+C_{e}}{3}$	$\frac{\alpha +C_{i}+C_{e}+mt}{3}$
Welfare of Country *E*	$\frac{\left(\alpha +C_{i}-2C_{e}\right)\left(\alpha +C_{i}-2C_{e}-3\varepsilon \right)}{9\beta }$	$\frac{{\left(\alpha +C_{i}-2C_{e}-2\varepsilon \right)}^{2}}{8\beta }$
Changes in the welfare of Country *E*	$\frac{\left(\alpha +C_{i}-2C_{e}\right)\left(\alpha +C_{i}-2C_{e}-12\varepsilon \right)+36\varepsilon^{2}}{72\beta }$	
Tariff revenue or subsidies	——	$\frac{\left(-\alpha -C_{i}+2C_{e}+6\varepsilon \right)\left(\alpha +C_{i}-2C_{e}-2\varepsilon \right)}{8\beta }$
Welfare of Country *I*	$\frac{{\left(\alpha +C_{e}-2C_{i}\right)}^{2}}{9\beta }+\frac{{\left(2\alpha -C_{e}-C_{i}\right)}^{2}}{18\beta }$ $+\varepsilon \frac{\alpha +C_{i}-2C_{e}}{3\beta }$	$\frac{{\left(\alpha -3C_{i}+2C_{e}+2\varepsilon \right)}^{2}}{16\beta }+\frac{{\left(3\alpha -C_{i}-2C_{e}-2\varepsilon \right)}^{2}}{32\beta }$ $+\frac{\varepsilon \left(3\alpha -C_{i}-2C_{e}-2\varepsilon \right)}{4\beta }$
Profit of Company *e*	$\frac{{\left(\alpha +C_{i}-2C_{e}\right)}^{2}}{9\beta }$	$\frac{{\left(\alpha +C_{i}-2C_{e}-2\varepsilon \right)}^{2}}{4\beta }$
Changes in profit of Company *e*	$\frac{\left(5\alpha +5C_{i}-10C_{e}-6\varepsilon \right)\left(\alpha +C_{i}-2C_{e}-6\varepsilon \right)}{6\beta }$	
Profit of company *i*	$\frac{{\left(\alpha +C_{e}-2C_{i}\right)}^{2}}{9\beta }$	$\frac{{\left(\alpha -3C_{i}+2C_{e}+2\varepsilon \right)}^{2}}{16\beta }$
Changes in profit of Company *i*	$\frac{\left(7\alpha -17C_{i}+10C_{e}+6\varepsilon \right)\left(-\alpha -C_{i}+2C_{e}+6\varepsilon \right)}{12\beta }$	

Note: BCAs is the abbreviation for border carbon adjustments. α represents the limit price when the supply of harvested wood products (HWPs) tends to zero; β represents the sensitivity of HWPs prices to supply; $C_{x}\left(x=e,i\right)$ is the marginal production cost of Company *x* (x=e or i); and ε represents the value of carbon storage.

## Data Availability

The datasets used or analyzed during the current study are available from the corresponding author on reasonable request.

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
