# Peer review of "Welfare Implications of Border Carbon Adjustments on the Trade of Harvested Wood Products"

_ijerph, 2022, doi:10.3390/ijerph20010790_

Round 1

Reviewer 1 Report

This manuscript concentrates on a topical policy instrument in the context of trade and climate change, the Border carbon adjustment (BCA)

Introduction:

line 25/26: Border carbon adjustment (BCA).

Overall well written and interesting related topics are mentioned, but it could be here and there even expanded, better explained as not all readesr are environmental economists. Also at the end the reserach focus could be more focused. I think it is difficult to "compare the changes in the behavior of governments and companies in the scenario with and without BCAs", as you wrote.

Methods: are you citing all used literature and methodological sources? For example in line 147 you write, that you use the "inverse solution method", like it is widely know, but there and also later I could not find a source in your text where it was applied in for example similar context/way. Also a kind graphics showing roles and flows between the two conutries would faciliate things for the reader. ou have there many assumptions and limitations mentioned. Instead of writing them just down you could make this clear in such a graphic, maybe in a map with the two conutries or list these things in a table. Then it will be more clear for the reader.

Results:

for me this is partly still methology (the many formulas and conditions and limitations of your model)

Discussion:

too short and also not sufficiently leaving your abstract model. I would expect here more linkages to reality , maybe also a flwo chart and accoring to model results + rleevance for reality two or maybe even three subchapters. If you want to stay with your tehoretical model I would put more emphasis on the practical side.

It seem there are quite strong limitations in your model (e.g. one company in each country only) that question for me the relevance of the results in the practical sense. Please consider that the Int. J. Environ. Res. Public Health  is a high level scientific journal that demands that the presented topic is as much as possible explored. From my perspective at least more linkages to practise, a better structered and larger discussion (e.g. reaching stronger out to policies and implcations on international level, e.g. global trade) and in general a better embedment into the Carbon context is needed. So I recommended major revision.

Author Response

All responses are attached.

Reviewer 2 Report

I liked the paper very much.  Here are a few major and minor comments,

Major comments

1.      Explain briefly on line 30 why BCAs shift the economic burden of emission reduction to non-emission reducing countries.

2.      Line 38, I don’t quite know what you mean by $50/ton were imposed on carbon.  Is this a tax?

3.      Line 90 – levy taxes on exported or import HWPs?  Perhaps you mean subsidize exports?

4.      Sentence on line 95 does not make sense because economic benefits and carbon stock values are not parallel?

5.      It looks to me that the countries are playing a Cournot game.  If so, please state it as such.  If not explain how the countries compete in the game process section.

6.      Remind the reader why the profit functions in equation 6 and 7 differ for countries I and E.

7.      Line 222 you state that equation 12 is for Country WE’s profit function, but it is shown as its welfare function.

8.      I don’t understand what you are saying in line 224.  If you take the derivative of welfare with respect to t and set the derivative =0 (first order condition), you can solve for the optimal t.  Is that what was done?

9.      How is tariff rate pass-through addressed?  Do you assume a 100% pass- through?  Please specify.

Minor

1.      Last sentence in the abstract does not make sense.  It looks like some words word dropped after effectively.

2.      Line 45 needs a transition.  You abruptly got to game theory.

3.      Line 88 should be HWP not HWPs

4.      Change article to paper throughout the paper.

5.      Line 143, I think you mean imposing a tariff

6.      Line 185 and equation 6, define t in mtQ, is the tariff rate or subsidy depending on the sign?. 

7.      Line 213, state, Company I’s profit increases ….

Author Response

All responses are attached.

Reviewer 3 Report

Review report on Welfare implications of border carbon adjustments on the trade 2 of harvested wood products

I have gone through the manuscript and found it fit for the scope of this journal. The issues investigated are very topical, relevant and stand the potential of advancing knowledge in the extant literature. This is especially considering the evolving global attention on the impacts of trade-led growth. I provide the following comments for the authors to consider in further improving the study.

1.      The abstract looks good but it would be fine if the sample country is reflected in the abstract. Besides, the methodology adopted in carrying out the study is very important.

2.      The introduction is well written however, certain basics are lacking in it. I provided a few as follows; (i) the information provided about BCAs and HWPs do not reflect how both remain a researchable topic in the sample countries. (ii) The objectives of the study should be clearly stated.

3.      As it stands, the contribution of the study is not substantial enough to qualify for publication. Before contributions can be advanced, it is important to know what has been done then clarify how the current study expands the knowledge, affirms or refute the existing findings. The mention of gravity model is far from being contribution.

4.      The literature review is missing in the paper. Was it deliberate? If yes, substantial reasons must be given to justify that. Else, a section should be created for that. It is important that studies in 2022, 2021, and 2019 should be given priority since they provide the recent arguments.

5.      The empirical model is well stated. However, I recommend that authors state which study they follow in specifying the empirical model for the study.

6.      The methodology section fails to provide the theoretical foundation and hypotheses guiding the nexuses among the indicators.

7.      The discussion of the results should be linked to recent studies.

8.      Conclusion can be improved.

9.      The policy recommendations are weak in their present form. They can strengthen it by relating the findings to the recommends.

10.   Limitation and future research opportunity are missing. They should be considered.

Overall, the paper is full of potentials that can be evident if the comments above are addressed.

Author Response

All responses are attached.

Round 2

Reviewer 3 Report

The authors have substantially addressed all issues raised. Many thanks to the authors for paying attentions to the comments and satisfactorily addressing them. I hereby recommend their paper for acceptance.